# Resolvin D2 and Resolvin D1 Differentially Activate Protein Kinases to Counter-Regulate Histamine-Induced [Ca^2+^]_i_ Increase and Mucin Secretion in Conjunctival Goblet Cells

**DOI:** 10.3390/ijms23010141

**Published:** 2021-12-23

**Authors:** Menglu Yang, Nora Botten, Robin Hodges, Jeffrey Bair, Tor P. Utheim, Charles N. Serhan, Darlene A. Dartt

**Affiliations:** 1Massachusetts Eye and Ear, Department of Ophthalmology, Schepens Eye Research Institute, Harvard Medical School, Boston, MA 02114, USA; nora_botten@hotmail.com (N.B.); robin_hodges@meei.harvard.edu (R.H.); Jeffrey_bair@meei.harvard.edu (J.B.); darlene_dartt@meei.harvard.edu (D.A.D.); 2Department of Medical Biochemistry, Oslo University Hospital, 0316 Oslo, Norway; utheim2@gmail.com; 3Department of Plastic and Reconstructive Surgery, Oslo University Hospital, 0316 Oslo, Norway; 4Faculty of Medicine, Institute of Clinical Medicine, University of Oslo, 0316 Oslo, Norway; 5Center for Experimental Therapeutics and Reperfusion Injury, Department of Anesthesiology, Perioperative and Pain Medicine, Brigham and Women’s Hospital and Harvard Medical School, Boston, MA 02115, USA; cserhan@bwh.harvard.edu

**Keywords:** ocular allergy, conjunctivitis, histamine, SPMs, RvD1, RvD2

## Abstract

Resolvin (Rv) D2 and RvD1 are biosynthesized from docosahexaenoic acid (DHA) and promote resolution of inflammation in multiple organs and tissues, including the conjunctiva. Histamine is a mediator produced by mast cells in the conjunctiva during the allergic response. We determined the interaction of RvD2 with histamine and its receptor subtypes in cultured conjunctival goblet cells and compared them with RvD1 by measuring intracellular [Ca^2+^] and mucous secretion. Treatment with RvD2 significantly blocked the histamine-induced [Ca^2+^]_i_ increase as well as secretion. RvD2 and RvD1 counter-regulate different histamine receptor subtypes. RvD2 inhibited the increase in [Ca^2+^]_i_ induced by the activation of H1, H3, or H4 receptors, whereas RvD1 inhibited H1 and H3 receptors. RvD2 and RvD1 also activate distinct receptor-specific protein kinases to counter-regulate the histamine receptors, probably by phosphorylation. Thus, our data suggest that the counter-regulation of H receptor subtypes by RvD2 and RvD1 to inhibit mucin secretion are separately regulated.

## 1. Introduction

Allergic conjunctivitis is a predisposition to hypersensitivity reactions upon exposure to specific environmental antigens and is characterized by ocular symptoms, such as itching, blurry vision, redness, and watery eyes, which are frequently combined with nasal and skin symptoms [1,2]. This type of conjunctivitis has an estimated prevalence of 15–40% worldwide [2,3]. All types of allergic conjunctivitis involve IgE-mediated type 1 hypersensitivity [4,5,6], in which IgE triggers the degranulation of mast cells. The granule content, mainly histamine and other immunoregulatory mediators [7], causes vasodilatation, increased capillary permeability, excessive mucin secretion [8], and itching [4,6]. Histamine also has an immunomodulatory effect that results in recurrence and chronic allergic conjunctivitis [9]. Currently, acute allergic conjunctivitis is treated with antihistamines or mast cell stabilizers. Both medications can effectively relieve ocular symptoms, but topical antihistamine therapy has a shorter duration and causes rebound hyperemia with continued use; topical mast cell stabilizers have a longer therapeutic effect but require a long loading period usually of days to weeks.

The conjunctiva, a target tissue of ocular allergy, is a mucous membrane that covers the outer surface of the eye and surrounds the cornea. The conjunctival goblet cell (CGC) is one of the major cell types found in the conjunctival epithelium [10,11]. CGCs synthesize, store, and secrete the high molecular weight glycoprotein mucin MUC5AC, along with other glycoproteins, electrolytes, and water, into the tear film [11]. This secretion forms the innermost layer of the tear film, which moisturizes, lubricates, and protects the ocular surface from exogenous pathogens and allergens in the non-diseased state. During ocular allergy, the CGCs are directly activated by inflammatory mediators, such as histamine and leukotrienes, which cause over secretion of mucin and lead to discomfort [8]. The four histamine receptor subtypes (H1–H4) have been observed on both human and rat CGCs and activation of each receptor triggers secretion [9]. Generally, the four histamine receptor subtypes activate different signaling pathways: (1) H1 is coupled to Gαq and increases intracellular [Ca^2+^]_i_ ([Ca^2+^]_i_) via phospholipase C (PLC) [9], which induces vasodilation and bronchoconstriction in allergies; (2) H2 is a Gαs coupled-receptor, activates the cAMP-PKA pathway [12], and is best known for mediating gastric acid release onto the gastric epithelium [13]; (3) H3 is Gαi/o coupled [14] and is functional in CGCs [9,15]; (4) H4 also couples to Gαi/o and is mainly distributed in lymphoid-related tissue, acting as an immune modulator [16,17]. Most antihistamine medications are targeted at H1 and/or H2 receptors. Even though H3 and H4 are equally important to histamine-induced mucin secretion on CGCs [9], no H3 or H4 modulator has been developed for clinical use. 

The D-series resolvins, RvD1-RvD6, are specialized pro-resolving mediators (SPMs) biosynthesized from ω-3-fatty acids initially discovered during the resolution phase of inflammation via a lipid mediator class switch [18] (Figure 1). RvD1 and RvD2 play a role in both health and disease. RvD1 and 2 are both present in human tears [18]. RvD1 activates ALX/FPR2 receptor in rat- and GPR32 in human-derived CGCs and induces mucin secretion via PLC and Ca^2+^ [19]. In contrast, RvD2 activates GPR18 in both rat- and human-derived CGCs and induces mucin secretion via PKA and Ca^2+^ [20]. Both RvDs enhance resolution during various types of inflammatory diseases, notably ocular allergy [21]. We observed the regulatory action of RvD1 and aspirin-triggered RvD1 (AT-RvD1) on histamine-induced CGC secretion [22]. Furthermore, topical RvD1 treatment was effective in decreasing the ocular allergy symptoms, including increased tear MUC5AC in a mouse model [23]. The mechanism used by RvD1 to block histamine stimulation of goblet cell secretion activated by the H1 receptor is to activate the β adrenergic receptor kinase 1 (βARK1) and protein kinase C (PKC) that probably phosphorylate the receptor at specific sites, causing the receptor to terminate its activity [22]. In the present study, we used cultured rat- and human-derived CGCs to determine which histamine receptor subtypes RvD2 and RvD1 counter-regulate and identify the protein kinases used.

## 2. Results

### 2.1. RvD2 Counter-Regulates Histamine-Stimulated Secretion from Rat- and Human-Derived Cultured Conjunctival Goblet Cells

To determine the regulatory action of RvD2 on histamine-stimulated secretions, CGCs derived from rats and humans were incubated with increasing concentrations of RvD2 (10^−9^–10^−7^ M) for 30 min before stimulation with histamine 10^−5^ M (Figure 2). In rat-derived CGCs, histamine alone significantly increased glycoprotein secretion by 2.0 ± 0.2-fold above basal (*p* = 0.004) (Figure 2a). The histamine-stimulated glycoprotein secretion in rat-derived cells was significantly inhibited to 1.5 ± 0.05-fold with RvD2 at 10^−9^ M (*p* = 0.048), 1.4 ± 0.07 with RvD2 at 10^−8^ M (*p* = 0.02) and 1.1 ± 0.01 with RvD2 at 10^−7^ M (*p* = 0.005). In human-derived cells, histamine alone significantly stimulated glycoprotein secretion by 2.1 ± 0.05-fold above basal (*p* < 0.001) (Figure 2b). Histamine-stimulated glycoprotein secretion in human cells was significantly inhibited by 1.5 ± 0.05 fold using RvD2 at 10^−9^ M (*p* = 0.001), 1.4 ± 0.07 using RvD2 at 10^−8^ M (*p* = 0.001) and 1.2 ± 0.02 using RvD2 at 10^−7^ M (*p* < 0.001). These findings indicate that RvD2 inhibits the histamine-induced secretion in CGCs.

### 2.2. RvD2 Counter-Regulates Histamine-Stimulated Increase in [Ca^2+^]_i_ in Cultured Conjunctival Goblet Cells Derived from Rats and Humans

To determine the action of RvD2 on the histamine-stimulated increase in [Ca^2+^]_i_, cultured CGCs derived from both rats and humans were incubated with RvD2 (10^−10^ M–10^−8^ M) for 30 min before histamine 10^−5^ M was added (Figure 3a). In CGCs derived from rats, histamine increased [Ca^2+^]_i_ in individual cells shown by pseudocolor images with blue indicating the lowest and red the highest [Ca^2+^]_i_ (Figure 3b). The [Ca^2+^]_i_ was measured before and after histamine stimulation (Figure 3c,d [cells derived from rats]; Figure 4a,b [cells derived from humans]), and the maximum change of [Ca^2+^]_i_ was calculated (Figure 3d and Figure 4b). The increase in [Ca^2+^]_i_ over time from all cells and experiments added together shows that the histamine-induced increase in [Ca^2+^]_i_ was decreased by each concentration of RvD2 used (Figure 3d). Figure 3c,d show that histamine significantly increased peak [Ca^2+^]_i_ by 484.0 ± 79.0 nM above basal (*p* < 0.001). Treatment with RvD2 (10^−10^ M) did not alter the histamine-stimulated increase in [Ca^2+^]_i_, while treatment with RvD2 (10^−9^ and 10^−8^ M) significantly blocked the histamine-stimulated increase in [Ca^2+^]_i_ to 147.5 ± 96.8 nM (*p* = 0.02) and 157.8 ± 43.4 nM (*p* = 0.02), respectively. Similar results were obtained in cells derived from humans as in those derived from rats. Histamine significantly increased peak [Ca^2+^]_i_ by 330.3 ± 47.4 nM above basal (*p* < 0.001) (Figure 4a,b). Treatment with RvD2 10^−10^–10^−9^ M did not block the histamine-stimulated increase in [Ca^2+^]_i_, while treatment with RvD2 10^−8^ M significantly decreased the histamine-stimulated increase in [Ca^2+^]_i_ to 185.1 ± 22.3 nM (*p* = 0.03). 

Our published data showed that RvD1 counter-regulates histamine response in human-derived CGCs [22]. We treated human CGCs with RvD1 10^−10^–10^−8^ M before the addition of histamine at 10^−5^ M (Figure 4c,d). Treatment with RvD1 10^−10^–10^−9^ M did not block the histamine-stimulated increase in [Ca^2+^]_i_, while treatment with RvD1 10^−8^ M significantly attenuated the histamine-stimulated increase in [Ca^2+^]_i_ to 122.5 ± 41.5 nM (*p* = 0.007). No significant difference was obtained with the level of inhibition between RvD1 and RvD2 10^−8^ M in human cells. In summary, both RvD2 and RvD1 inhibit histamine-induced [Ca^2+^]_i_ increases in both rat- and human-derived CGCs.

### 2.3. RvD2 Uses PKA to Counter-Regulate Histamine-Stimulated Increase in [Ca^2+^]_i_ in Cultured Conjunctival Goblet Cells Derived from Rats

The phosphorylation sites for individual protein kinases on histamine receptor subtypes were predicted using *Scansite4*, where protein kinase A (PKA) phosphorylation sites were predicted on all four histamine receptor subtypes. To determine if RvD2 uses PKA to counter-regulate the histamine-stimulated increase in [Ca^2+^]_i_, PKA inhibitor H89 (10^−5^ M) was given 15 min prior to RvD2 treatment (Figure 5a,b). Histamine at 10^−5^ M alone increased peak [Ca^2+^]_i_ by 384.4 ± 87.6 nM above basal (*p* = 0.001). Treatment with RvD2 at 10^−8^ M significantly decreased the histamine-stimulated increase in [Ca^2+^]_i_ to 127.8 ± 25.3 nM (*p* = 0.02). Treatment with H89 10^−5^ M alone did not alter the histamine-stimulated response. The counter-regulation by RvD2 to the histamine-stimulated response was successfully reversed by treatment with H89 (10^−5^ M) to 262.5 ± 46.2 nM. This finding indicates that RvD2 uses PKA to counter-regulate the histamine receptor-stimulated increase in [Ca^2+^]_i_. 

### 2.4. RvD2 Counter-Regulates Increase in [Ca^2+^]_i_ Stimulated by Histamine Receptor 1, 3, and 4 Specific Agonists, while RvD1 Counter-Regulates Response to Histamine Receptor 1 and 3 Specific Agonists in Cultured Conjunctival Goblet Cells Derived from Rats

To determine the histamine receptor subtypes that RvD2 regulates, cultured rat CGCs were preincubated with RvD2 10^−8^ M for 30 min before agonists specific to the histamine receptors H1-H4 were added individually (Figure 6). Histamine dimaleate (H1 agonist) at 10^−6^ M significantly increased peak [Ca^2+^]_i_ by 302.0 ± 56.8 nM above basal (*p* < 0.001) (Figure 6a,b). Treatment with RvD2 10^−8^ M significantly decreased the H1-stimulated increase in [Ca^2+^]_i_ to 133.6 ± 22.6 nM (*p* = 0.02). Amthamine dihydrobromide (H2 agonist) at 10^−5^ M increased peak [Ca^2+^]_i_ by 345.7 ± 49.7 nM above basal (*p* < 0.001). Treatment with RvD2 10^−8^ M did not alter the H2-stimulated increase in [Ca^2+^]_i_ (Figure 6a,b) (*p* = 0.3). (R)-(-)-α-methylhistamine (H3 agonist) at 10^−5^ M increased peak [Ca^2+^]_i_ by 416.5 ± 50.1 nM above basal (*p* = 0.001, Figure 6c,d). Treatment with RvD2 10^−8^ M significantly blocked the H3-stimulated increase in [Ca^2+^]_i_ to 106.9 ± 14.3 nM (*p* < 0.001). 4-methylhistamine dihydrochloride (H4 agonist) at 10^−5^ significantly increased peak [Ca^2+^]_i_ by 421.6 ± 73.6 nM above basal (*p* < 0.001) (Figure 6c,d). Treatment with RvD2 10^−8^ M significantly blocked the H4-stimulated increase in [Ca^2+^]_i_ to 156.1 ± 26.7 nM (*p* = 0.003). As a positive control, histamine-stimulated increase in [Ca^2+^]_i_ was significantly downregulated by RvD2 treatment (*p* < 0.001) (Figure 6a,d).

A similar experiment was conducted using RvD1 to compare its counter-regulatory function on histamine receptor subtypes with that of RvD2 (Figure 6e,g). The H1 agonist was the positive control [22]. Figure 6e,g demonstrates that the H1 agonist at 10^−6^ M significantly increased peak [Ca^2+^]_i_ by 198.7 ± 23.5 nM above basal (*p* < 0.001), which is consistent with our previous findings [22]. Treatment with RvD1 at 10^−8^ M significantly decreased the H1-stimulated increase in [Ca^2+^]_i_ to 123.2 ± 10.4 nM (*p*= 0.01). H2 agonist at 10^−5^ M increased peak [Ca^2+^]_i_ by 259.7 ± 28.4 nM above basal (*p* < 0.001). Treatment with RvD1 at 10^−8^ M did not alter the H2-stimulated increase in [Ca^2+^]_i_ (*p =* 0.07). The H3 agonist at 10^−5^ M increased peak [Ca^2+^]_i_ by 282.9 ± 53.1 nM above basal (*p* < 0.001). Treatment with RvD1 at 10^−8^ M significantly blocked the H3-stimulated increase in [Ca^2+^]_i_ to 141.4 ± 14.7 nM (*p* = 0.03). The H4 agonist at 10^−5^ M increased peak [Ca^2+^]_i_ by 212.8 ± 33.6 nM above basal, while RvD1 at 10^−8^ M did not significantly alter the H4-stimulated increase in [Ca^2+^]_i_ (*p =* 0.1).

In summary, RvD2 and RvD1 each counter-regulates the action of histamine via different histamine receptor subtypes with RvD2 acting on H1, 3, and 4 and RvD1 acting on H1 and 3. Neither resolvin counter-regulates the H2 receptor subtype.

### 2.5. RvD2 Uses β-ARK1 to Counter-Regulate Increase in [Ca^2+^]_i_ Induced by H1 and H4, but Not H3 Receptor Subtype Agonists in Cultured Conjunctival Goblet Cells Derived from Rats

β-ARK, later named G-protein-coupled receptor kinase, was initially described as phosphorylating β-adrenergic receptors and promoting the interaction of the receptor with β-arrestin to counter-regulate the activation of G-proteins and associated signaling pathways [25]. We published that RvD1 and arachidonic acid-derived specialized lipid mediator lipoxin A_4_ (LXA_4_) activate β-ARK1 to counter-regulate the action of the H1 receptor subtype agonist, histamine dimaleate, on CGCs to increase [Ca^2+^]_i_ [22,26]. To determine if RvD2 uses β-ARK1 to counter-regulate the action of H1, H3, and H4 receptors, β-ARK1 inhibitor (β-ARK1I, 10^−6^ M) was given 15 min before H receptor subtype agonist or RvD2 (10^−8^ M) administration. RvD2 stimulation was followed 30 min later by H receptor subtype agonist. The H1 agonist at 10^−6^ M increased [Ca^2+^]_i_ by 320.9 ± 47.4 nM above basal (*p* < 0.001) (Figure 7a,b). RvD2 pre-treatment significantly decreased the H1-induced [Ca^2+^]_i_ increase to 121.1 ± 11.3 nM (*p =* 0.01). β-ARK1I treatment did not alter stimulation by H1 agonist (a control for a direct effect on the H1 receptor subtype), but reversed the inhibitory action of RvD2 on H1 agonist. These results indicate that RvD2 activates β-ARK1 to downregulate the H1 receptor subtype. 

When CGCs were stimulated by the H3 agonist, RvD2 successfully downregulated the increase in [Ca^2+^]_i_ induced by the H3 agonist at 10^−6^ M (*p =* 0.01) (Figure 7c,d). β-ARK1I treatment did not alter stimulation by the H3 agonist. In contrast to the H1 agonist, β-ARK1I treatment did not reverse the inhibitory action of RvD2 on the H3 agonist (*p* = 0.01).

A result similar to that of the H1 agonist was obtained with the H4 agonist at 10^−5^ M. RvD2 treatment significantly decreased the H4 agonist-induced increase in [Ca^2+^]_i_ (Figure 7e,f). β-ARK1I treatment did not alter stimulation by H4 agonist, but β-ARK1I treatment reversed the inhibitory effect of RvD2 on the H4 agonist (*p* = 0.3) 

These results indicate that RvD2 activates β-ARK1 to downregulate the H1 and H4, but not the H3, receptor subtypes.

### 2.6. RvD1 Uses β-ARK1 to Counter-Regulate the Increase in [Ca^2+^]_i_ Induced by H3 Receptor Subtype in Cultured Conjunctival Goblet Cells Derived from Rats

Our published data showed that RvD1 uses β-ARK1 to counter-regulate activity of H1 receptor subtype [22]. We determined whether RvD1 counter-regulated the H3 receptor subtype. The H3 agonist at 10^−6^ M increased [Ca^2+^]_i_ in cultured rat CGCs by 255.0 ± 36.4 nM (*p* < 0.001) (Figure 8a,b). Pre-treatment of RvD1 at 10^−8^ M for 30 min significantly attenutated the increase in [Ca^2+^]_i_ induced by the H3 agonist to 98.2 ± 10.7 nM (*p* = 0.003). β-ARKI treatment alone did not alter the H3-triggered [Ca^2+^]_i_ increase. β-ARKI 15 min treatment successfully reversed the downregulation by RvD1 on H3 to 262.8 ± 62.5 nM (*p* = 0.9) (Figure 8a,b). These results indicate that RvD1 activates β-ARK1 to downregulate the H1 and H3 receptor subtypes.

### 2.7. RvD2 Uses PKA to Counter-Regulate the Increase in [Ca^2+^]_i_ Induced by H1 and H3 Receptor Subtypes Agonists, but Uses PKC for the Action of H4 Receptor Subtype Agonist in Cultured Conjunctival Goblet Cells Derived from Rats

As *Scansite4* predicted PKA and PKC phosphorylation sites on histamine receptors, the PKA inhibitor H89 at 10^−5^ M or the PKC inhibitor Ro-317549 at 10^−7^ M were used in rat CGCs, 15 min prior to histamine receptor subtype agonists or RvD2 (10^−8^ M) treatment that was then followed 30 min later by stimulation by individual histamine receptor subtype agonists. Activation of the H1 receptor by its agonist at 10^−6^ M increased [Ca^2+^]_i_ by 435.4 ± 33.2 nM above basal (*p* < 0.001) (Figure 9a,b). Treatment with RvD2 significantly blocked the H1 agonist-stimulated increase in [Ca^2+^]_i_ to 119.1 ± 15.2 nM (*p* < 0.001). Neither H89 nor Ro-317549 treatment altered stimulation by the H1 agonist. Preincubation with H89 successfully reversed the inhibition of H1 response by RvD2. In contrast, the inhibition caused by RvD2 was not reversed by Ro-317549 (Figure 9a,b).

Results similar to H1 response were obtained with the H3 agonist. The H3 agonist at 10^−6^ M increased [Ca^2+^]_i_ by 255.5 ± 38.9 nM above basal (*p* < 0.001) (Figure 9c,d). Prior treatment with RvD2 (10^−8^ M) significantly blocked the H3-stimulated increase in [Ca^2+^]_i_ to 111.6 ± 19.4 nM (*p =* 0.01). Neither H89 nor Ro-317549 treatment altered stimulation by the H3 agonist. Treatment with H89 before RvD2 and then H3 agonist addition reversed the inhibition of H3 response by RvD2. In contrast, the inhibition of H3 caused by RvD2 was not reversed by Ro-317549 (Figure 9c,d). 

H89 and Ro-317549 were next studied on the H4 response. The H4 agonist at 10^−5^ M increased [Ca^2+^]_i_ by 300.0 ± 66.9 nM above basal (*p* < 0.001) (Figure 9e,f) Prior incubation with RvD2 (10^−8^ M) significantly downregulated the H4-stimulated increase in [Ca^2+^]_i_ to 90.4 ± 22.5 nM (*p =* 0.01). Neither H89 nor Ro-317549 treatment altered stimulation by the H4 agonist. H89 15 min pretreatment did not reverse the inhibition of H4 response by RvD2. In contrast, the inhibition caused by RvD2 was reversed by Ro-317549 (Figure 9e,f). 

These results indicate that RvD2 activated PKA, but not PKC, to counter-regulate the increase in [Ca^2+^]_i_ induced by H1 and H3 receptor activation. In contrast, RvD2 activated PKC, but not PKA, to counter-regulate the H4 response. 

### 2.8. RvD1 Uses Both PKA and PKC to Counter-Regulate the Increase in [Ca^2+^]_i_ Induced by H1 and H3 Agonists in Cultured Conjunctival Goblet Cells Derived from Rats 

To compare with activation of PKA and PKC by RvD2 to counter-regulate histamine receptor subtypes, similar experiments were performed using RvD1 (Figure 10a–f). The H1 agonist at 10^−6^ M increased [Ca^2+^]_i_ by 268.2 ± 47.8 nM above the basal (*p* < 0.001) (Figure 10a–c). Prior incubation with RvD1 (10^−8^ M) significantly blocked the H1 agonist-stimulated increase in [Ca^2+^]_i_ to 114.9 ± 10.9 nM (*p =* 0.01). Neither H89 nor Ro-317549 treatment altered stimulation by the H1 agonist. Treatment with H89 successfully reversed the inhibition of H1 response by RvD1 (Figure 10a,c). The inhibition caused by RvD1 was also reversed by Ro-317549 (Figure 10b,c). 

Similar results were obtained with H3 agonist, which increased [Ca^2+^]_i_ by 229.7 ± 32.3 nM above basal (*p* < 0.001) (Figure 10d–f). Prior incubation with RvD1 10^−8^ M significantly blocked the H3-stimulated increase in [Ca^2+^]_i_ to 98.9 ± 8.8 nM (*p =* 0.002). Neither the H89 nor Ro-317549 treatment altered stimulation by the H3 agonist. Addition of H89 or Ro-317549, respectively, reversed the inhibition of H3 response by RvD1 to 205.8 ± 55.7 and 176.1 ± 20.4 nM, respectively, (Figure 9d–f). These results indicate that RvD1 activated both PKA and PKC to counter-regulate the [Ca^2+^]_i_ increase induced by the H1 and H3 receptor subtype activation. 

## 3. Discussion

In the present study, we present evidence that RvD2 and RvD1 both inhibit histamine-induced increase in [Ca^2+^]_i_ and mucin secretion in CGCs by acting via different histamine receptor subtypes. The action of histamine via H1 and H3 receptors is inhibited by both RvDs, while the H4 receptor is only attenuated by RvD2. Neither RvD2 nor RvD1 blocks the action of the H2 receptor. Each receptor was modulated by phosphorylation by different kinases, while β-ARK1 was generally relatively used by both RvDs to counter-regulate histamine receptors (Table 1, Figure 11). 

Histamine and its role in allergies has been extensively described. Despite the differences in signaling pathways and physiological functions, histamine receptor subtypes work synergistically to induce and maintain allergic diseases. The number and type of H receptor subtypes vary between cells and tissues. H1 receptors induce vasodilation and cause edema. H2 and 4 receptors are expressed on eosinophils [17] and mast cells [27]. H4 induces chemotaxis of eosinophils and mast cells caused by histamine and enhances the effect of other chemotactic agents [17,27]. H1 and H4 receptors on dendritic cells activate and induce T-helper (Th)2 polarization, inducing inflammation to become chronic [28,29]. New evidence shows that H2 receptors on monocytes also induce Th2 cell polarization [30]. Additionally, H1 and H4 receptor levels increase on the ocular surface during chronic allergic conjunctivitis [10]. In contrast to H1 and H4 receptors, activation of H2 receptors inhibits histamine release and along with H3 receptors modulates cytokine production from mast cells [31]. H2 also weakens histamine-induced chemotaxis in eosinophils [32]. Taken together, this evidence shows that H1 and H4 play a major role in inducing allergy and inflammation, while H2 negatively regulates this process. Herein, both RvD1 and RvD2 attenuated H1 receptor activation on conjunctival goblet cells, and neither of these RvDs altered H2 receptors. These results indicate that both RvD1 and RvD2 are potential anti-allergic agents. In addition, RvD2 attenuates H4 activity, indicating RvD2 may be more effective in attenuating allergic disease than RvD1. 

In the present study, we observed that RvD2 and RvD1 use different protein kinases to modulate the same H receptor subtype (Figure 11). Among the kinases we investigated in the current study, β-ARK is involved in the counter-regulation of multiple receptors [33,34]. β-ARK1 binds to the beta-gamma subunit of the G protein, phosphorylates the C-terminal tail of GPCRs, and then recruits β-arrestin to downregulate the receptor and internalize it [33,34]. β-ARK1 thus attenuates or alters the signaling pathways activated by the receptor. This mechanism makes β-ARK a common regulator of G-protein signaling pathways, and the role of β-ARK in regulating H1 [35], H2 [36] and H4 [37] receptors has been published. We conclude that in conjunctival goblet cells RvD1 and RvD2 using their respective cognate receptor mediated signaling pathways activate β-ARK1 counter-regulating H1/H3/H4 and blocking their stimulated increase in [Ca^2+^]_i_ and hence their stimulation of secretion (Figure 11c,d). We propose that this counter-regulation is facilitated through the recruitment of β-arrestin by the activated β-ARK1 to phosphorylate and internalise the histamine receptor subtypes. In contrast to the other histamine receptor subtypes, the H2 receptor is not counter-regulated by either RvD1 or RvD2 and could continue signaling when activated. In addition to being counter-regulated by β-ARK1, the H1, H3, and H4 receptors are regulated by other protein kinases, which we suggest are dependent on the protein kinase interaction with specific phosphorylation sites identified in G protein-coupled receptors including the four histamine receptor subtypes. According to the *Scansite4* prediction, the PKA phosphorylation site on H1 and H3 receptor subtypes is on the second intracellular loop, whereas the PKC site is on the third loop. For the H4 receptor subtype, both PKA and PKC phosphorylation sites are located on the third intracellular loop. If the different phosphorylation sites or events are not equal, this could explain why RvD2-H1 and -H3 interactions only involve PKA, but not PKC. Similarly this suggestion could explain why RvD2-H4 uses only uses PKC, but not PKA. In contrast to RvD2, RvD1 regulates H1 and H3 receptors by modulating both the PKC and PKA phosphorylation sites on these histamine receptor subtypes. In addition to the histamine receptor subtype-specific phosphorylation sites, a second variable is important for the type of phosphorylation site used. That variable is the type of kinases activated by RvD1 and RvD2. This is more complicated, however, than just the use of different protein kinases. In a separate study we found that RvD1 activates PKC, but not PKA, to increase the [Ca^2+^]_i_. In contrast, RvD2 activates PKA, but not PKC, to increase [Ca^2+^]_i_. Thus the protein kinases used by RvD1 and RvD2 to counter-regulate the histamine receptor subtypes are different than those used to activate phospholipase C, D, and A2 to increase [Ca^2+^]_i_ and activate PKC or adenylyl cyclase to increase cAMP/PKA. The difference could be due to the PKC or PKA isoform used, cellular localization of the process (compartmentalization), or timing (30-min pre-treatment for counter-regulation action compared to immediate for resolvin treatment alone). These distinct activities could explain why RvD1 and RvD2 and other SPMs used alone stimulate secretion, but when used before the addition of a pro-inflammatory mediator inhibit its action and block secretion. These dissimilar effects of RvD1 and RvD2 could also explain how SPMs can function in maintaining the mucous layer in health by stimulating secretion and returning the overproduction of mucins to homeostasis in inflammatory diseases by inhibiting secretion.

RvD2 is biosynthesized from one of the ω3-fatty acids docosahexaenoic acid (DHA) that is converted to 17S-hydroxy-4Z,7Z,10Z,13Z,15E,19Z-DHA (17S-HpDHA) by 15-lipoxygenase type I (15-LOX). 17S-HpDHA next transforms to 7(8) epoxide-containing intermediate by 5-lipoxygenase (LOX) this then converted into both RvD1 or RvD2 [24]. Both RvD1 and RvD2 were identified in human tears [18]. The RvD2-GPR18 axis exists and functions in multiple systems. Among functional studies, RvD2 shares some actions with RvD1 in physiological functions, such as anti-inflammation [38], anti-depression [13], vascular regulation [39], and analgesia [40]. There are some differences in potency between these two RvDs. RvD2 has higher potency in inhibiting inflammatory pain than RvD1 [40], while in the current study, there were no statistically significant differences in inhibitory potency found between RvD1 and RvD2 in histamine-triggered CGC activation.

In health as compared to disease, RvD1 and RvD2 stimulate CGC secretion, helping to maintain a normal mucin amount in the tear film. On CGCs, despite the similarities in synthesis and functions of RvD2 and RvD1, there are differences in receptors used and intracellular signaling pathways between RvD2 and RvD1 to directly stimulate goblet cell function, as could occur in health. RvD2 activates the GPR18 receptor in both CGCs derived from both humans and rats. RvD2 elevates the cAMP level via the activation of PKA, then causes [Ca^2+^]_i_ increase by interacting with inositol trisphosphate (IP_3_) receptor on the ER leading to mucin secretion [20]. In contrast, RvD1 uses ALX/FPR2 receptors in CGCs derived from rats and GPR32 in CGC derived from humans. RvD1 activates PLC, which generates IP_3_ and diacylglycerol (DAG), leading to an increase in [Ca^2+^]_i_ and mucin secretion; RvD1 also activates PLA_2_ and PLD, that result in the activation of extracellular regulated kinase (ERK)1/2 [19]. In inflammatory disease, such as ocular allergy, these differences in intracellular signaling pathways used by RvD1 and RvD2 may not explain the distinct histamine receptor subtypes interacted with by the two RvDs to counter-regulate histamine signals, as RvD1 uses PKA to counter-regulate H1 and H3 receptor subtypes as in disease, but does not use PKA directly to stimulate secretion, as could occur in health. Additionally, the H2 receptor subtype signals by PKA as in health, but RvD2 activation of PKA that stimulates secretion directly, does not counter-regulate the H2 receptor subtype as in disease. Thus, counter-regulation of H receptor subtypes by RvDs in ocular allergy to inhibit mucin secretion and RvD-induced direct stimulation of CGC mucin secretion in health are separately regulated.

Rat conjunctival goblet cells extensively resemble human conjunctival goblet cells. In the present study, we observed the downregulation of histamine-triggered [Ca^2+^]_i_ increase by RvD2 in both human and rat CGCs. There is an 84.89% similarity in the RvD2 receptor GPR18 sequence between humans and rats, according to the Ensembl database [41]. Although RvD1 uses a different receptor in human tissues than in the rat, we observed that human and rat goblet cells are similar in their RvD1, aspirin-triggered RvD1 and histamine-induced [Ca^2+^]_i_, increase [22]. Thus, conjunctival goblet cells cultured from rats are an accurate model for conjunctival goblet cells cultured from humans.

In conclusion, RvD2 and RvD1 differentially counter-regulate histamine receptor subtypes by activating distinct receptor-specific protein kinases to downregulate the histamine receptor subtype cellular signaling pathway. This mechanism provides new potential targets for the treatment of conjunctival allergic disease. Moreover, the counter-regulation of H receptor subtypes by specific resolvins in ocular allergy to inhibit mucin secretion and resolvin-induced direct stimulation of CGC mucin secretion in health, are distinctly regulated.

## 4. Materials and Methods

### 4.1. Materials

Roswell Park Memorial Institute (RPMI) 1640 cell culture medium, penicillin/streptomycin, and l-glutamine were purchased from Lonza Group (Basel, Switzerland). Fetal bovine serum was from Atlanta Biologicals (Flowery Branch, GA, USA). Fura-2- acetoxymethyl ester and Amplex Red were from Thermo Fisher Scientific (Waltham, MA, USA). Pluronic acid F127, sulfinpyrazone, and carbachol (Cch) were from MilliporeSigma (Burlington, MA, USA). RvD2 (7S,16R,17S-trihydroxy-4Z,8E,10Z,12E,14E,19Z-docosahexaenoic acid) and RvD1 (7S,8R,17S-trihydroxy-4Z,9E,11E,13Z,15E,19Z-docosahexaenoic acid) were purchased from Cayman Chemical Company (Ann Arbor, MI, USA). 

N-[2-(p-bromocinnamylamino)ethyl]-5-isoquinolinesulfonamide dihydrochloride (H89) was purchased from MilliporeSigma (Burlington, MA, USA). Ro 317549 (3-(1-(3-Aminopropyl)-1H-indol-3-yl)-4-(1-methyl-1H-indol-3-yl)-1H-pyrrole-2,5-dione acetate monohydrate) was purchased from Tocris Bioscience (Minneapolis, MN, USA). β-ARK1 inhibitor (Methyl 5-[(E)-2-(5-nitrofuran-2-yl)ethenyl]furan-2-carboxylate) was purchased from Santa Cruz Biotechnology (Santa Cruz, CA, USA). 

Agonists and inhibitors were diluted in RPMI 1640 medium for secretion experiments and Krebs-Ringer bicarbonate buffer with 4-(2-hydroxyethyl)-1-piperazineethanesulfonic acid [KRB-HEPES; 119 mM NaCl, 4.8 mM KCl, 1.0 mM CaCl_2_, 1.2 mM MgSO_4_, 1.2 mM KH_2_PO_4_, 25 mM NaHCO_3_, 10 mM HEPES, and 5.5 mM glucose (pH 7.45)] for [Ca^2+^]_i_ measurements. 

Scansite (https://scansite4.mit.edu/4.0/#home, accessed on 8 October 2021) was used to determine phosphorylation sites of receptors.

### 4.2. Animals

Conjunctiva removed from four- to eight-week-old male Sprague-Dawley rats (Taconic Biosciences, Rensselaer, NY, USA) was used. Experiments were performed according to the ARVO Statement for the Use of Animals in Ophthalmic and Vision Research and approved by Schepens Eye Research Institute Animal Care and Use Committee. 

### 4.3. Human Tissue

Male and female conjunctiva was obtained from Eversight (Ann Arbor, MI, USA). Tissue was placed in Opsitol GS media within 18 h after death. Use of this tissue was reviewed by the Massachusetts Eye and Ear Human Studies Committee and determined to be exempt and does not meet the definition of research with human subjects.

### 4.4. Cell Culture

Rat and human CGCs were grown from explants in cell culture. CGCs were grown in 6-well plates containing RPMI 1640 medium supplemented with 10% bovine serum, 2 mM glutamine and 100 µg/mL penicillin-streptomycin. To ensure that goblet cells predominate in the cultures, cells were regularly stained with Helix pomatia agglutinin (HPA, human cells) or Ulex europaeus agglutinin-1 (UEA-1, rat cells) to detect goblet cell secretory product, as well as cytokeratin 7 to detect intermediate filaments.

### 4.5. High-Molecular Weight Glycoprotein Secretion Measurements

Goblet cells were passaged, seeded into 24-well plates, and grown to 75% confluence. Thereafter, cells were serum-starved for 2 h in serum-free RPMI 1640 medium supplemented with 0.5% bovine serum albumin, incubated with inhibitors for 30 min, and stimulated with agonists for 4 h. To measure secretion, an enzyme-linked lectin assay (ELLA) with horseradish peroxidase-conjugated UEA-1 was used. The medium was collected and transferred to Nunc microplates (Thermo Fisher Scientific). A standard curve was generated using bovine submaxillary mucin. The standards and medium were dried overnight at 60 °C and UEA-1 was detected using Amplex Red and quantified using a fluorescence ELISA reader (model Synergy MX; BioTek Instruments, Winooski, VT, USA) with excitation and emission wavelengths of 530 and 590 nm, respectively. Total protein was determined using the Bradford assay. Secretion was normalized to total protein and expressed as fold increase above the basal that was set to 1. This assay measures UEA1-detectable glycoproteins that include the mucin MUC5AC, the major goblet cell secretory mucin, as goblet cells release all of their secretory granules upon stimulation [42]. 

### 4.6. [Ca^2+^]_i_ Measurements

Goblet cells were passaged and seeded onto glass-bottom dishes. Cells were incubated at 37 °C with KRB-HEPES containing 0.5% bovine serum albumin, 0.5 μM Fura-2/AM, 8 μM pluronic acid F127, and 250 μM sulfinpyrazone for 1 h. Before use, cells were washed in KRB-HEPES with sulfinpyrazone. A ratio imaging system (In Cyt Im2; Intracellular Imaging, Cincinnati, OH, USA) with excitation wavelengths of 340 and 380 nm and an emission wavelength of 505 nm was used to measure [Ca^2+^]_i_. Inhibitors were added for 15 min and resolvins 30 min before stimulation with agonists. Data are shown as the actual [Ca^2+^]_i_ with time or as the change in peak [Ca^2+^]_i_, calculated by subtracting the average of the basal value from the peak [Ca^2+^]_i_.

### 4.7. Statistical Analysis

The data are presented as average ± SEM. Student’s *t*-test was used for comparison of two groups and one-way ANOVA with Dunnett’s multiple comparisons was used to perform statistical analysis in multiple groups. *p* < 0.05 was set as statistically significant.

## Figures and Tables

**Figure 1 ijms-23-00141-f001:**
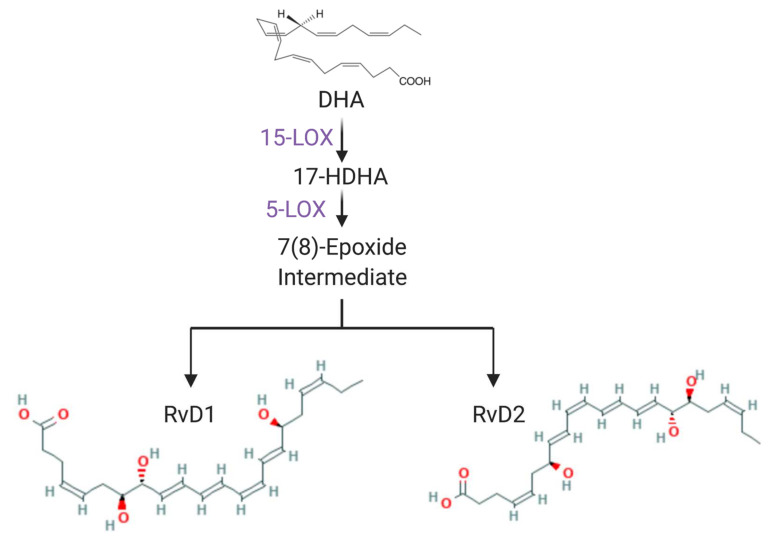
The biosynthesis of RvD1 and RvD2. The docosahexaenoic acid (DHA) is one of the ω-3-fatty acids [24]. DHA is first converted into 17-hydroxy docosahexaenoic acid (17-HDHA) by the 15-lipoxygenase (15-LOX), and further converted into the 7(8)-epoxide intermediate, and finally converted to 7*S*,8*R*,17*S*-trihydroxy-DHA (RvD1) or 7*S*,16*R*,17*S*-trihydroxy-DHA (RvD2). The structure of RvD2 is acquired from the National Center for Biotechnology Information, Resolvin D1 (CID44251266), Resolvin D2 (CID 11383310). The oxygen atom is marked in red.

**Figure 2 ijms-23-00141-f002:**
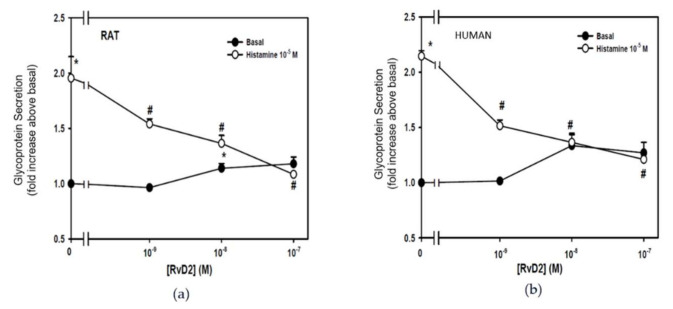
RvD2 counter-regulates histamine-stimulated high molecular weight glycoprotein secretion in cultured conjunctival goblet cells derived from both rats and humans. CGCs derived from rat (**a**) and human (**b**) were incubated with increasing concentrations of RvD2 (10^−9^–10^−7^ M) for 30 min before stimulation with histamine 10^−5^ M and glycoprotein secretion was measured. The average fold increase above basal was calculated and shown in (**a**) for rat-derived and (**b**) for human-derived cells. Data are mean ± SEM from 3 rats and 3 humans. * Indicates a significant difference from basal. # Indicates a significant difference from stimulus alone.

**Figure 3 ijms-23-00141-f003:**
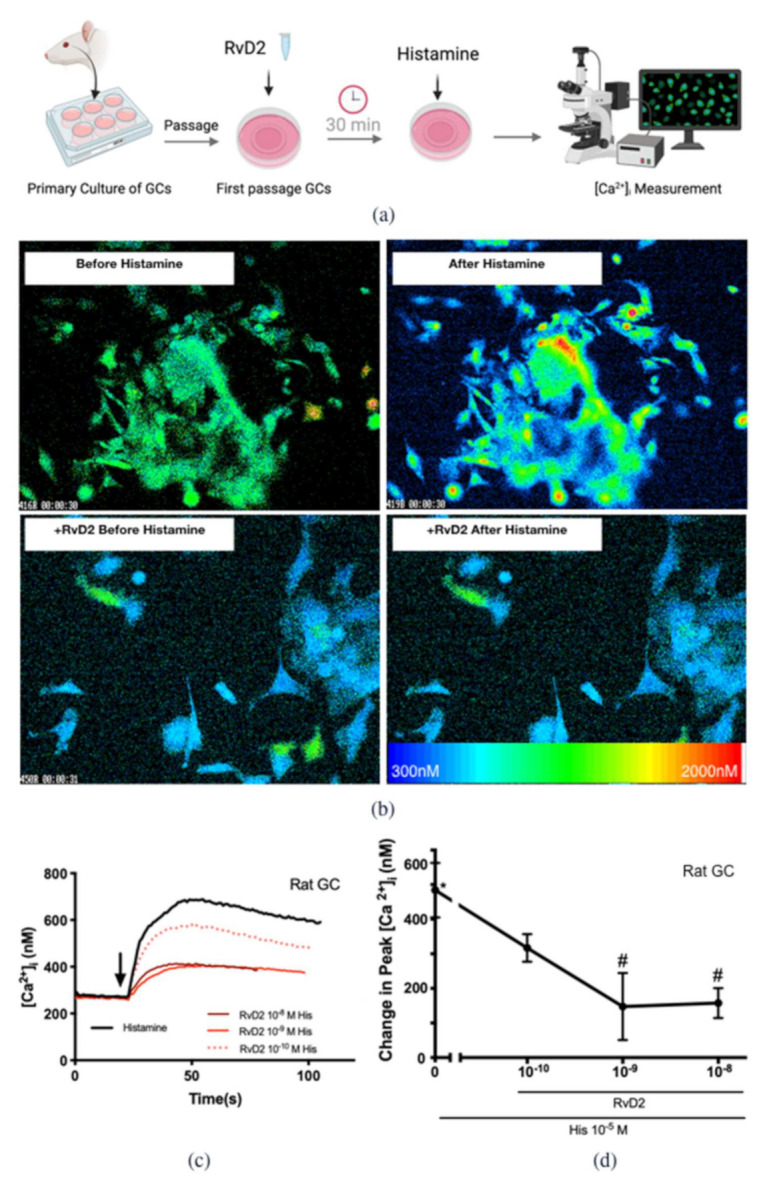
RvD2 counter-regulates histamine-stimulated increase in [Ca^2+^]_i_ in cultured conjunctival goblet cells derived from rats. Rat goblet cells were cultured from conjunctival tissue for 1 week, and then passaged to the glass-bottom dishes one day prior to the experiment. The goblet cells were preincubated with RvD2 (10^−10^–10^−8^ M) or vehicle for 30 min prior to the addition of histamine (10^−5^ M) and [Ca^2+^]_i_ was measured as illustrated in (**a**). Pseudocolor micrographs of [Ca^2+^]_i_ signal in CGCs before the addition of histamine (left panels) and after histamine (right panels) is shown in (**b**). The upper panels represent control experiments and the lower panels represent experiments with the addition of RvD2. The spectrum on the lower panel indicates the [Ca^2+^]_i_ concentration. The average [Ca^2+^]_i_ over time was shown in (**c**); and change in peak [Ca^2+^]_i_ was calculated and shown in (**d**). Data are mean ± SEM from 5 rats. * Indicates a significant difference from basal. # Indicates a significant difference from stimulus alone. Arrow indicates addition of histamine.

**Figure 4 ijms-23-00141-f004:**
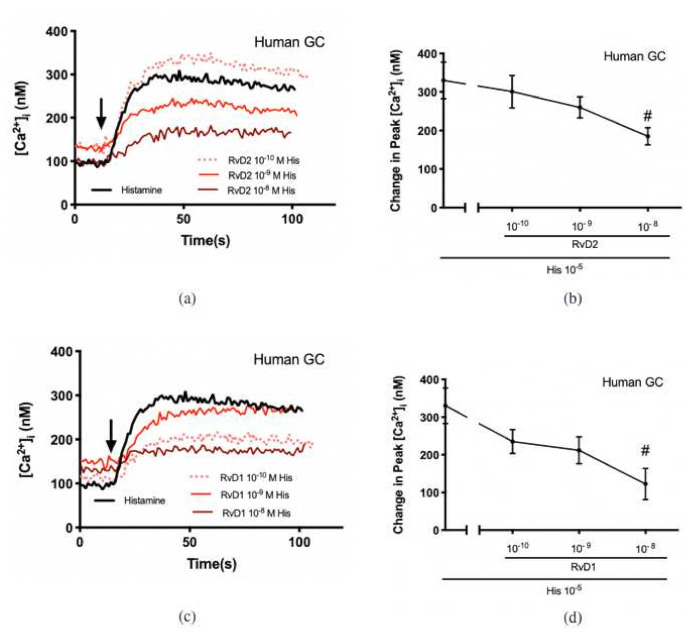
RvD2 counter-regulates histamine-stimulated increase in [Ca^2+^]_i_ in cultured conjunctival goblet cells derived from humans. Human goblet cells were preincubated with RvD2 (10^−10^–10^−8^ M) (**a**,**b**), RvD1 (10^−10^–10^−8^ M) (**c**,**d**), or vehicle for 30 min prior to the addition of histamine (10^−5^ M), and [Ca^2+^]_i_ was measured. The average [Ca^2+^]_i_ level over time was shown in (**a**,**c**); and change in peak [Ca^2+^]_i_ was calculated and shown in (**b**,**c**). Data are mean ± SEM from 4 humans. # Indicates a significant difference from stimulus alone. Arrow indicates addition of histamine.

**Figure 5 ijms-23-00141-f005:**
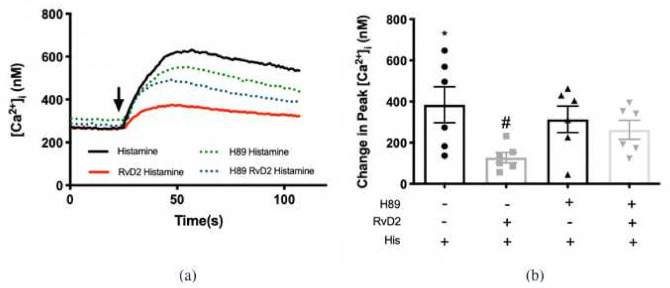
RvD2 uses PKA to counter-regulate histamine-stimulated increase in [Ca^2+^]_i_. Histamine (10^−5^ M) was added alone (solid black line in a; first bar in b). RvD2 (10^−8^ M) was added followed by histamine 30 min later (red line in a; second bar in b). PKA inhibitor H89 (10^−5^ M) was given 30 min prior to addition of histamine (dotted green line in a; third bar in b). PKA inhibitor H89 was given 15 min prior to RvD2 (10^−8^ M) treatment followed 30 min later by histamine (10^−5^ M) (dotted blue line in a; fourth bar in b). The average [Ca^2+^]_i_ level over time was shown in (**a**); change in peak [Ca^2+^]_i_ was calculated and shown in (**b**). Data are mean ± SEM from 6 rats. * Indicates a significant difference from basal. # Indicates a significant difference from stimulus alone. Arrow indicates addition of histamine.

**Figure 6 ijms-23-00141-f006:**
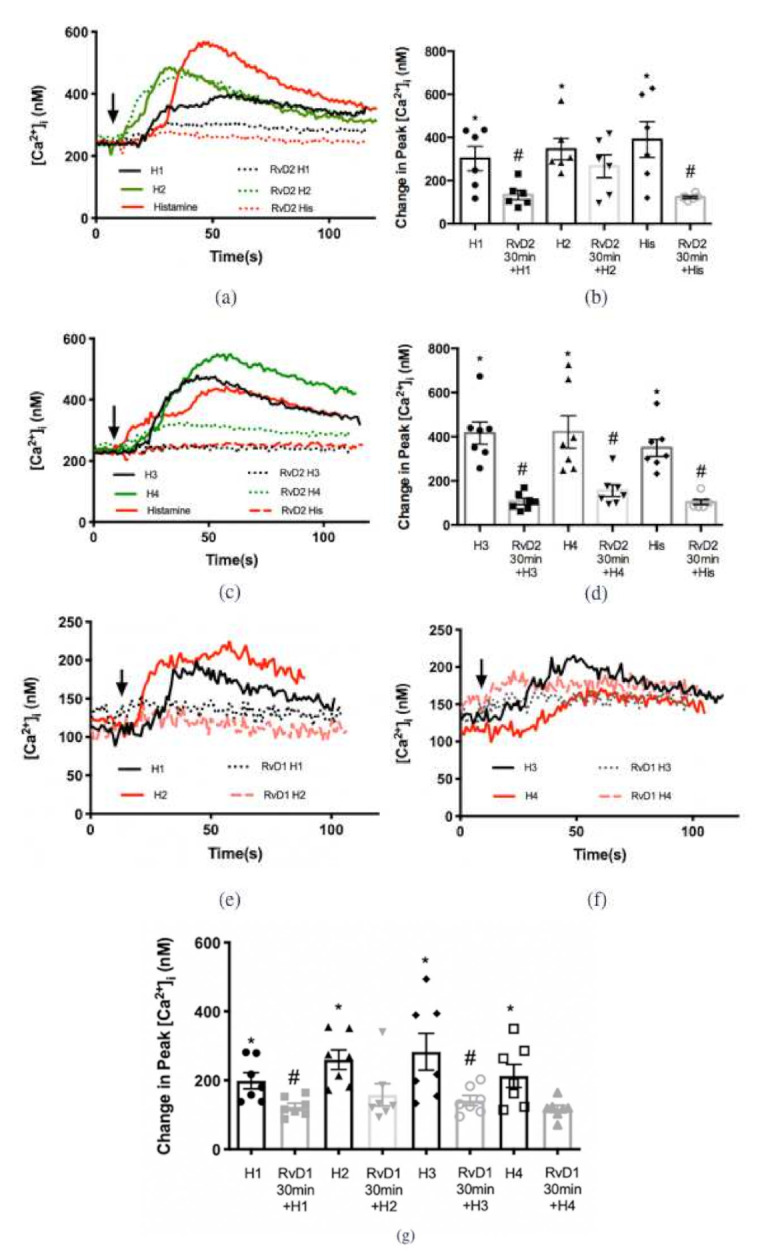
RvD2 counter-regulates the increase in [Ca^2+^]_i_ stimulated by histamine receptor 1, 3, and 4 specific agonists, while RvD1 inhibits histamine receptor 1 and 3 specific agonists. Rat goblet cells were treated with an H1 agonist (histamine dimaleate, 10^−6^ M), H2 agonist (amthamine dihydrobromide, 10^−5^ M) (**a**,**b**, bars 1 and 3), H3 agonist ((R)-(-)-α-methylhistamine, 10^−5^ M), H4 agonist (4-methylhistamine dihydrochloride,10^−5^ M) (**c**,**d**, bars 1 and 3) or histamine (10^−5^ M) (**a**–**d**, bar 5). Cells were also pretreated with RvD2 (10^−8^ M) for 30 min prior to the addition of H1 (10^−6^ M), H2 (10^−5^ M) (**a**,**b**, bars 2 and 4), H3 (10^−5^ M), H4 (10^−5^ M) (**c**,**d**, bars 2 and 4) specific agonists or histamine (10^−5^ M) (**a**–**d**, bar 6) respectively. Rat goblet cells were treated with an H1 agonist (histamine dimaleate, 10^−6^ M), H2 agonist (amthamine dihydrobromide, 10^−5^ M) (e and g), H3 agonist ((R)-(-)-α-methylhistamine, 10^−5^ M), H4 agonist (4-methylhistamine dihydrochloride, 10^−5^ M) (**f**,**g**, bars 1, 3, 5 and 7, respectively). Cells were also pretreated with RvD1 (10^−8^ M) (**e**–**g**) for 30 min prior to the addition of H1 (10^−6^ M), H2 (10^−5^ M) (**e**,**g**), H3 (10^−5^ M), H4 (10^−5^ M) (**f**,**g**, bars 2, 4, 6, and 8, respectively) specific agonists. The average [Ca^2+^]_i_ over time was shown in (**a**,**c**,**e**,**f**); change in peak [Ca^2+^]_i_ was calculated and shown in (**b**,**d**,**g**). Data are mean ± SEM from 6 rats. * Indicates a significant difference from basal. # Indicates a significant difference from stimulus alone. Arrow indicates addition of histamine or histamine receptor subtype agonists.

**Figure 7 ijms-23-00141-f007:**
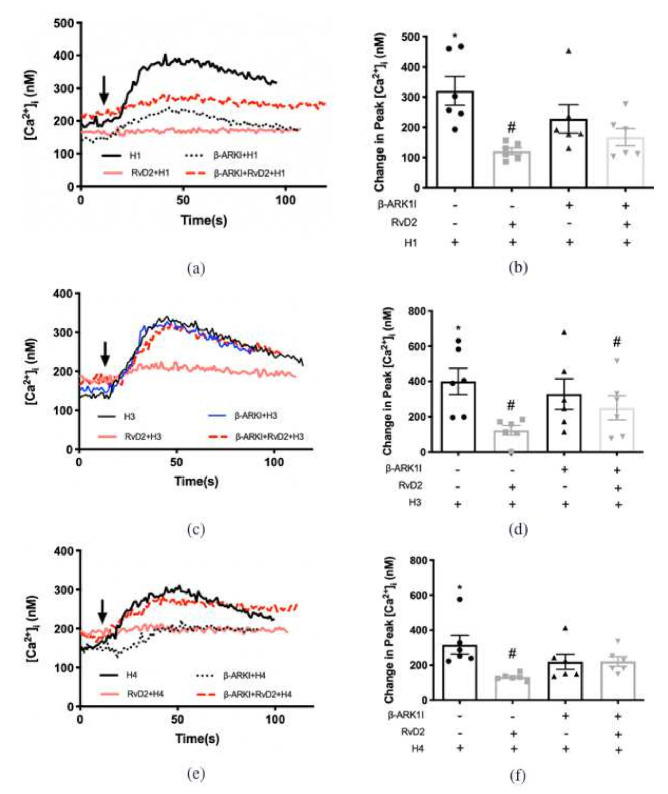
RvD2 uses β-ARK1 to counter-regulate the increase in [Ca^2+^]_i_ induced by H1 and H4 specific agonists, but not the H3 specific agonist. H1 agonist (histamine dimaleate, 10^−6^ M) (**a**,**b**), H3 agonist ((R)-(-)-α-methylhistamine, 10^−5^ M) (**c**,**d**), and H4 agonist (4-methylhistamine dihydrochloride, 10^−5^ M) (**e**,**f**) were added alone (solid black line in **a**,**c**,**e**; first bar in **b**,**d**,**f**). RvD2 (10^−8^ M) was added followed by an H1 (**a**,**b**), H3 (b and d) or H4 (**e**,**f**) agonist 30 min later (solid pink line in **a**,**c**,**e**; second bar in **b**,**d**,**f**). β-ARK1 inhibitor β-ARK1I (10^−6^ M) was given 30 min prior to addition of H1 (**a**,**b**), H3 (**b**,**d**) or H4 (**e**,**f**) agonists (dotted black line in **a**,**c**,**e**; third bar in **b**,**d**,**f**). β-ARK1 inhibitor β-ARK1I (10^−6^ M) was given 15 min prior to RvD2 (10^−8^ M) treatment followed 30 min later by H1 (**a**,**b**), H3 (**c**,**d**) or H4 (**e**,**f**) agonist (dotted red line in **a**,**c**,**e**; fourth bar in **b**,**d**,**f**). The average [Ca^2+^]_i_ level over time was shown in (**a**,**c**,**e**); change in peak [Ca^2+^]_i_ was calculated and shown in (**b**,**d**,**f**). Data are mean ± SEM from 6 rats. * Indicates a significant difference from basal. # Indicates a significant difference from stimulus alone. Arrow indicates addition of histamine receptor subtype agonist.

**Figure 8 ijms-23-00141-f008:**
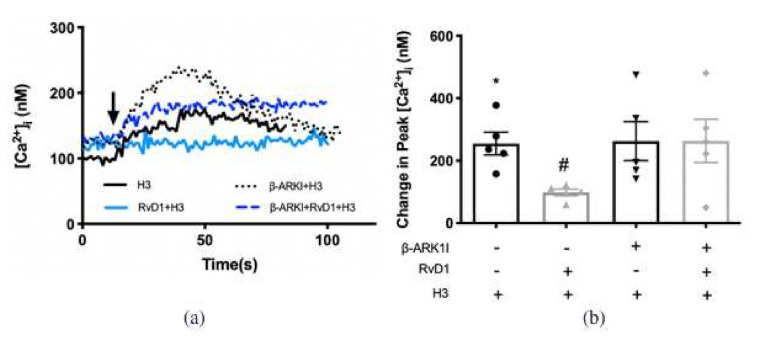
RvD1 uses β-ARK1 to counter-regulate increase in [Ca^2+^]_i_ induced by H3 agonist. H3 agonist ((R)-(-)-α-methylhistamine, 10^−5^ M) was added alone (solid black line in a; first bar in **b**). RvD1 (10^−8^ M) was added followed by H3 agonist 30 min later (turquoise line in **a**; second bar in **b**). β-ARK1 inhibitor β-ARK1I (10^−6^ M) was given 30 min prior to addition of H3 agonist (dotted black line in **a**; third bar in **b**). β-ARK1 inhibitor β-ARK1I (10^−6^ M) was given 15 min prior to RvD2 (10^−8^ M) treatment followed 30 min later by H3 agonist (blue line in **a**; fourth bar in **b**). The average [Ca^2+^]_i_ level over time was shown in (**a**); change in peak [Ca^2+^]_i_ was calculated and shown in (**b**). Data are mean ± SEM from 5 rats. * Indicates a significant difference from basal. # Indicates a significant difference from stimulus alone. Arrow indicates addition of H3 receptor agonist.

**Figure 9 ijms-23-00141-f009:**
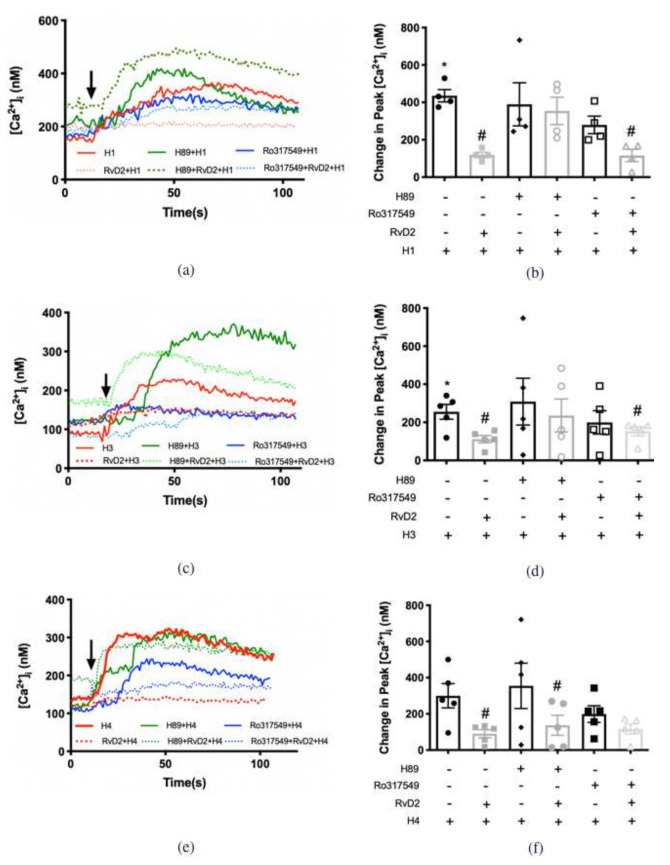
RvD2 uses PKA to counter-regulate the increase in [Ca^2+^]_i_ induced by H1 and H3 specific agonists, but uses PKC to counter-regulate the H4 specific agonist. H1 agonist (histamine dimaleate, 10^−6^ M) (**a**,**b**), H3 ((R)-(-)-α-methylhistamine, 10^−5^ M) (**c**,**d**), and H4 agonist (4-methylhistamine dihydrochloride, 10^−5^ M) (**e**,**f**) were added alone (solid red line in **a**,**c**,**e**; first bar in **b**,**d**,**f**). RvD2 (10^−8^ M) was added followed by an H1 (**a**,**b**), H3 (**c**,**d**) or H4 (**e**,**f**) agonist 30 min later (dotted red line in **a**,**c**,**e**; second bar in **b**,**d**,**f**). PKA inhibitor H89 (10^−5^ M) was given 30 min prior to addition of H1 (**a**,**b**), H3 (**b**,**d**) or H4 (**e**,**f**) agonist (solid green line in **a**,**c**,**e**; third bar in **b**,**d**,**f**). PKA inhibitor H89 (10^−5^ M) was given 15 min prior to RvD2 (10^−8^ M) treatment followed 30 min later by H1 (**a**,**b**), H3 (**c**,**d**) or H4 (**e**,**f**) agonist (dotted green line in **a**,**c**,**d**; fourth bar in **b**,**d**,**f**). PKC inhibitor Ro317549 (10^−7^ M) was given 30 min prior to addition of H1 (**a**,**b**), H3 (**c**,**d**) or H4 (**e**,**f**) agonist (solid blue line in **a**,**c**,**e**; fifth bar in **b**,**d**,**f**). PKC inhibitor Ro317549 (10^−7^ M) was given 15 min prior to RvD2 (10^−8^ M) treatment followed 30 min later by H1 (**a**,**b**), H3 (**c**,**d**) or H4 (**e**,**f**) agonist (dotted blue line in **a**,**c**,**e**; sixth bar in **b**,**d**,**f**). The average [Ca^2+^]_i_ over time was shown in (**a**,**c**,**e**); change in peak [Ca^2+^]_i_ was calculated and shown in (**b**,**d**,**f**). Data are mean ± SEM from 4 rats for (**a**,**b**), 6 rats for (**c**–**f**). * Indicates a significant difference from basal. # Indicates a significant difference from stimulus alone. Arrow indicates addition of histamine receptor subtype agonist.

**Figure 10 ijms-23-00141-f010:**
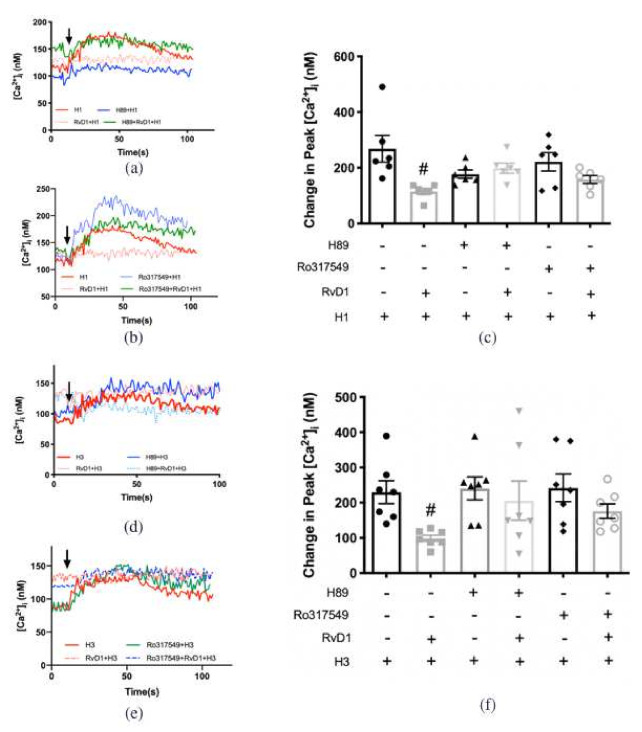
RvD1 uses both PKA and PKC to counter-regulate the increase in [Ca^2+^]_i_ induced by H1 and H3 specific agonists. H1 agonist (histamine dimaleate,10^−6^ M) (**a**,**b**) and H3 agonist ((R)-(-)-α-methylhistamine, 10^−5^ M) (**c**,**d**), were added alone (solid red line in **a**,**c**; first bar in **b**,**d**). RvD1 (10^−8^ M) was added followed by H1 (**a**,**b**) or H3 (**c**,**d**) agonist 30 min later (solid pink line in **a**,**c**; second bar in **b**,**d**). PKA inhibitor H89 (10^−5^ M) was given 30 min prior to addition of H1 (**a**,**b**) or H3 (**c**,**d**) agonist (blue line in **a**,**c**; third bar in **b**,**d**). PKA inhibitor H89 (10^−5^ M) was given 15 min prior to RvD1 (10^−8^ M) treatment followed 30 min later by H1 (**a**,**b**) or H3 (**c**,**d**) agonist (dashed blue line in **a**,**c**; fourth bar in **b**,**d**). PKC inhibitor Ro317549 (10^−7^ M) was given 30 min prior to addition of H1 (**a**,**b**) or H3 (**c**,**d**) agonists (green line in **a**,**c**; fifth bar in **b**,**d**). PKC inhibitor Ro317549 (10^−7^ M) was given 15 min prior to RvD1 (10^−8^ M) treatment followed 30 min later by H1 (**a**–**c**) or H3 (**d**–**f**) agonist (dashed blue line in **a**,**b**; sixth bar in **b**,**d**). The average [Ca^2+^]_i_ level over time was shown in (**a**,**b**,**d**,**e**); change in peak [Ca^2+^]_i_ was calculated and shown in (**c**,**f**). Data are mean ± SEM from 6 rats for (**a**–**c**), 7 rats for (**d**–**f**). # Indicates a significant difference from stimulus alone. Arrow indicates addition of histamine receptor subtype agonist.

**Figure 11 ijms-23-00141-f011:**
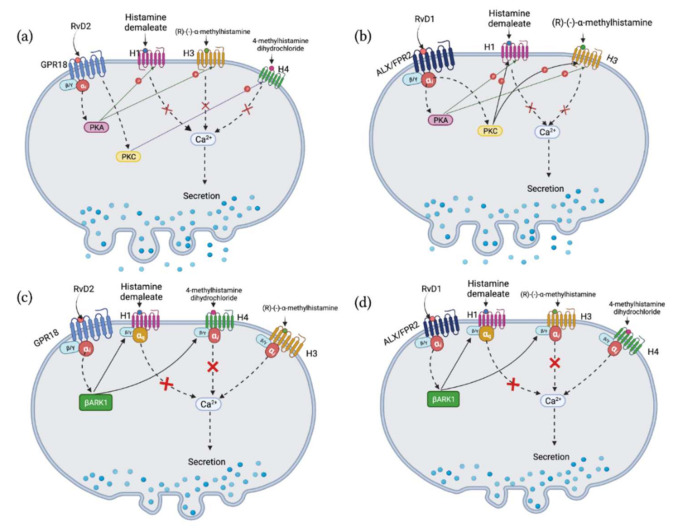
Summary of protein kinases used by RvD2 and RvD1 to counter-regulate increase on [Ca^2+^]_i_ induced by histamine receptor subtype agonists. RvD2 counter-regulates histamine-induced [Ca^2+^]_i_ increase by interaction with H1, H3, and H4 receptors (**a**,**c**). RvD2 uses PKA to counter-regulate [Ca^2+^]_i_ increase induced by H1 and H3 specific agonists, but uses PKC to regulate the H4 specific agonist (**a**). RvD2 Uses β-ARK1 to counter-regulate [Ca^2+^]_i_ increase induced by H1 and H4 specific agonists, but not H3 specific agonists (**c**). RvD1 counter-regulates histamine-induced increase in [Ca^2+^]_i_ by interaction with H1 and H3 (**b**,**d**). RvD1 uses PKA and PKC (**b**) and β-ARK1 (**d**) to counter-regulate increase in [Ca^2+^]_i_ induced by both H1 and H3 specific agonists. Solid lines with arrowheads indicate direct interaction; dash lines with arrowheads indicate indirect interaction; letter P in a circle indicates phosphorylation; and red crosses on the lines indicate this pathway is inhibited. Abbreviations: H1–H4, histamine receptor subtype 1–4; histamine dimaleate, H1 agonist; (R)-(-)-α-methylhistamine, H3 agonist; 4-methylhistamine dihydrochloride, H4 agonist; RvD, resolvin D; PKA, protein kinase A; PKC, protein kinase C; β-ARK1, β adrenergic receptor kinase 1; GPR18, G protein coupled receptor 18; ALX/FPR2, G-protein coupled formyl peptide receptor 2.

**Table 1 ijms-23-00141-t001:** Summary of protein kinases activated by RvD1 and RvD2 to counter-regulate H1–4 receptor subtypes.

		RvD2			RvD1		
H Receptors	Pathway	β-ARK1	PKA	PKC	β-ARK1	PKA	PKC
H1	Gq-PKC-IP3	+	+	–	+	+	+
H2	Gαs—cAMP	Not inhibited by RvD2			Not inhibited by RvD1		
H3	Gαi/o	–	+	–	+	+	+
H4	Gαi/o	+	–	+	Not inhibited by RvD1		

H, histamine; RvD, resolvin D; PKC, protein kinase C; IP3, Inositol trisphosphate; β-ARK1, β adrenergic receptor kinase 1; PKA, protein kinase A; G, G protein coupled receptor.

## Data Availability

Raw data were generated at Schepens Eye Research Institute. Derived data supporting the findings of this study are available from the corresponding author M.Y. on request.

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
