# Peer review of "Resolvin D2 and Resolvin D1 Differentially Activate Protein Kinases to Counter-Regulate Histamine-Induced [Ca2+]i Increase and Mucin Secretion in Conjunctival Goblet Cells"

_ijms, 2021, doi:10.3390/ijms23010141_

Round 1

Reviewer 1 Report

In the study entitled “Resolvin D2 and Resolvin D1 Differentially Activate Protein Kinases to Counter-regulate Histamine-induced [Ca2] Increase and Mucin Secretion in Conjuncitval Goblet Cells” the authors performed a thorough investigation of the molecular mechanism responding to histamine insult in rat and human goblet derived cells, with a focus on its inhibition by RvD1 and RvD2. Their findings support the notion that RvD1 and RvD2 inhibit mucin secretion via different pathways, including various histamine receptors and downstream kinases. Their work presented is detailed and impactful. Appreciated is the examination of the process in two different species, although continuation of their experiments using human derived cells would have had an even bigger impact. There are some minor deficiencies that should be addressed to further strengthen the study, but overall, this is an appropriately completed study.

GENERAL comments:

  • The textual use of “rat” and “human” throughout out the MS needs to be more specific as to rat derived cells or human derived goblet cells.
  • The Ca2+ graphs throughout the MS would visually benefit from color in addition to the line differences.
  • The authors should mention why after figure 2 they concentrated their effort exclusively on the rat derived cells.
  • Line 197 – figure reference is missing that its figure 5,

SPECIFIC Details

  • Figure 2:
    1. it would be beneficial to show images of RvD2 induced inhibition of histamine-mediated Ca2+ response.
    2. Figure legend indicates panel (c), which is missing from the figure?

  • Figure 3:
    1. Why is the histamine response higher than control at the lowest RvD2 concentration in (a)?
    2. For the data for peak Ca2+, there is no indication of what the levels look like without any RvD treatment, this would be a nice contrast to the data indicating inhibition is occurring in the presence of RvD.

  • Figure 4:
    1. Why do the authors change from line graph to bar graph to depict peak Ca2+? Consistency throughout the MS would be beneficial for the readers. If the line graph version is chosen, please ensure that individual points are shown as they are in the bar graphs. In fig 4b we see peak Ca2+ has a wide range from below 200 to above 600, yet in Figure 2c the error bars indicate a much smaller deviation from the mean (also a much smaller mean). This inconsistency should be addressed and individual points on the graphs would help with that.
    2. Why are there major differences between the peak Ca2+ levels when comparing H1 and H2 alone in 4b vs 4d? The difference in the means is almost ½, can the authors please comment on this inconsistency? Similar patterns are seen for H3 and H4 treatments? The Ca2+ concentration over time graphs 4e/f are also far less clear on the induction and inhibition of Ca2+ levels compared to 4a and 4b?

  • Figure 9:
    1. The Ca2+ over time graphs in this figure appear to have data that is much more variable? Also graph 9a has an axis that includes the 0 value on the Y axis while the others do not and the total values on the Y axis are not consistent. Can the authors comment on this data and why the Ca2+ values appear to fluctuate so much over time in comparison to their previous graphs?

Author Response

Reviewer 1:

Comments and Suggestions for Authors

In the study entitled Resolvin D2 and Resolvin D1 Differentially Activate Protein Kinases to Counter-regulate Histamine-induced [Ca2] Increase and Mucin Secretion in Conjuncitval Goblet Cells” the authors performed a thorough investigation of the molecular mechanism responding to histamine insult in rat and human goblet derived cells, with a focus on its inhibition by RvD1 and RvD2. Their findings support the notion that RvD1 and RvD2 inhibit mucin secretion via different pathways, including various histamine receptors and downstream kinases. Their work presented is detailed and impactful. Appreciated is the examination of the process in two different species, although continuation of their experiments using human derived cells would have had an even bigger impact. There are some minor deficiencies that should be addressed to further strengthen the study, but overall, this is an appropriately completed study.

Response I-1: We thank Reviewer 1 for recognizing the merits and innovation of this work. We also appreciate the helpful suggestions to increase clarity and allow the most informative interpretation of data. Each suggestion has been addressed in the manuscript revision, as detailed below: 

GENERAL comments:

The textual use of rat” and human” throughout out the MS needs to be more specific as to rat derived cells or human derived goblet cells.

Response I-2: The changed the wording throughout the manuscript as suggested.

The Ca2+ graphs throughout the MS would visually benefit from color in addition to the line differences.

Response I-3: The Ca2+ traces are now changed to color.

The authors should mention why after figure 2 they concentrated their effort exclusively on the rat derived cells. 

Response I-4: We switched to rat derived GCs for the following reasons:

  1. The human conjunctival tissue used in the present study was from deceased donors obtained from an eye bank, thus the age and sex of the available human tissue is limited due. In contrast, using rats we can easily control age and sex. 
  2. Because the human tissue is from donors that are deceased, the age of the donors is usually advanced (65-85 years old). In the human conjunctiva, ageing, disease, and environmental exposure may have affected the conjunctival GCs and decreasing their growth in culture and acting on their responsiveness. Using rat derived cells allows us to investigate the signaling pathways from cells that are younger, grow faster in culture and respond more consistently.
  3. As addressed in the Discussion we have consistently found that rat derived GCs resemble human derived GCs. Results in our recently submitted manuscript also suggest that RvD2 signals the same way in GCs derived from humans and rats. Thus, we used rat derived GCs for more complicated experiments, and only use human GCs to show the inhibition of histamine response by RvD2.

Line 197 – figure reference is missing that its figure 5,

Response I-5:  The Reference has been added (now Figure 6).

SPECIFIC Details

Figure 2:

it would be beneficial to show images of RvD2 induced inhibition of histamine-mediated Ca2+ response.

Figure legend indicates panel (c), which is missing from the figure?

Response I-6: Figure 2 is now figure 3. The pseudocolor images with RvD2 treatment has been added. We show the pseudocolor images before addition of histamine (basal), after addition of histamine, RvD2 just before addition of histamine, and histamine addition after RvD2 treatment. The pseudocolor resembles the ratio of fluorescence signals of 2 wavelength (340 nm/380 nm), and we are comparing the change of the ratio before and after the addition of the agonist. Because the RvD2 was added 30 min before the agonist, we are not able to show the reduction of signal. We can only show that there is minimum change before and after the addition of histamine in RvD2 treated cells. We also modified the illustration of the experiment for a better understanding.

The figure has been reformatted. 

Figure 3:

Why is the histamine response higher than control at the lowest RvD2 concentration in (a)?

For the data for peak Ca2+, there is no indication of what the levels look like without any RvD treatment, this would be a nice contrast to the data indicating inhibition is occurring in the presence of RvD.

Response I-7: Figure 3 is now figure 4. The RvD2 (10-10M) with histamine trace looks higher because it has a higher baseline. As explained in Response I-6, we are analyzing the change between peak and basal. This can be seen better in the revised figure with color traces. 

The first data point is histamine without RvD2. We made it stand out more by adding a break in the axis and using a different symbol in the revised figure (see new Figure 4).

Figure 4:

Why do the authors change from line graph to bar graph to depict peak Ca2+? Consistency throughout the MS would be beneficial for the readers. If the line graph version is chosen, please ensure that individual points are shown as they are in the bar graphs. In fig 4b we see peak Ca2+ has a wide range from below 200 to above 600, yet in Figure 2c the error bars indicate a much smaller deviation from the mean (also a much smaller mean). This inconsistency should be addressed and individual points on the graphs would help with that.

Response I-8: Fig 4 is now fig 5. The line graph used in Figure 4  is the standard way of demonstrating the concentration  response. We believe using the line graph shows the relationship between concentrations on a number line. Unfortunately, we are not able to put individual data points on this type of graph due to technical difficulty, thus we used the error bars to indicate the variability. 

Thought the study, we used cells cultured from different animals or humans. The cultured cells are primary cells and not cell lines. Furthermore, each experimental condition is performed in a separate culture well. The individuals bring variability that we cannot control. From the data published by our group and other research groups, the variability with [Ca2+]i measurements is common. 

Why are there major differences between the peak Ca2+ levels when comparing H1 and H2 alone in 4b vs 4d? The difference in the means is almost ½, can the authors please comment on this inconsistency? Similar patterns are seen for H3 and H4 treatments? The Ca2+ concentration over time graphs 4e/f are also far less clear on the induction and inhibition of Ca2+ levels compared to 4a and 4b?

Response I-9: See Figure 6. As explained in Response I-8, the cells we used are not from a cell line but from individual animals. The animals are caged in groups of  3-4, and usually the animals within the same cage have similar genetic background. However, when experiments are conducted a few weeks apart, the cells from newly arrived animals may respond differently. This variability from the animals is beyond our control. 

 Figure 9:

The Ca2+ over time graphs in this figure appear to have data that is much more variable? Also graph 9a has an axis that includes the 0 value on the Y axis while the others do not and the total values on the Y axis are not consistent. Can the authors comment on this data and why the Ca2+ values appear to fluctuate so much over time in comparison to their previous graphs?

Response I-10: Fig 9 is now Fig 10. As explained before, the variability is caused by cells cultured from different animals. Moreover, [Ca2+] the responses in this graph are lower than in other graphs, thus amplifying the fluctuation of the traces. By color-coding each individual traces, we hope to make the results more understandable.

The Y axis has been changed in the revised figure as indicated by the reviewer (see new figure 10).

Reviewer 2 Report

This MS provides a comprehensive view on resolvins' action on the histamine-induced [Ca2+]i increase. While inhibitory action of resolvins on the histamine-stimulated [Ca2+]i increase was clearly presented, the role of protein kinases in the process is not clearly presented.

Some points should be clarified:

  1. Line 59-60

H4 also couples to Gαi/o, activates GTPγS. 

It is not clear what is implied, exactly

  1. A very inaccurate Fig. 2. Even with different a, b, and C styles. b and С for the same image?
  2. 5e shows RvD1 inhibition. While in the MS text line 195

Treatment with RvD1 at 10-8M did not alter the H2-stimulated increase in [Ca2+]i (p=0.07).

  1. All lines on Fig. 5f look very similar.
  2. No direct evidence of phosphorylation (activation) is provided. For example, experiments with [gamma-32P]-ATP may provide it.

Author Response

Reviewer 2:

Comments and Suggestions for Authors

This MS provides a comprehensive view on resolvins' action on the histamine-induced [Ca2+]i increase. While inhibitory action of resolvins on the histamine-stimulated [Ca2+]i increase was clearly presented, the role of protein kinases in the process is not clearly presented.

Response II-1: We thank the Reviewer for pointing out the need for clarification. Each suggestion has been addressed in the manuscript revision, as detailed below:

Line 59-60 H4 also couples to Gαi/o, activates GTPγS. It is not clear what is implied, exactly

Response II-2: According to Ref 15, “Although SP9144 (the H4 receptor) apparently coupled to Galpha(i), HEK-293 cells stably transfected with SP9144 did not exhibit histamine-mediated inhibition of forskolin-stimulated cAMP levels. However, both [(35)S] GTPgammaS binding and phosphorylation of mitogen-activated protein kinase were stimulated by histamine via SP9144 activation.” However, we realized that the sentence is confusion and is not related to our current study in which we measured the Ca2+ signal and not cAMP. Thus, we removed the comments on Gi/o. 

A very inaccurate Fig. 2. Even with different a, b, and C styles. b and С for the same image?

Response II-3: This figure (now Figure 3) has been re-formatted correctly. As requested by Reviewer 1, pseudocolor images of GCs with RvD2 treatment have been added. Please note that the pseudocolor indicates the ratio of 2 fluorescence signals (340 nm and 380nm) and the change of the ratio is compared.

5e shows RvD1 inhibition. While in the MS text line 195

Treatment with RvD1 at 10-8M did not alter the H2-stimulated increase in [Ca2+]i (p=0.07).

Response II-4:  Fig 5 is now Fig 6. The purpose of the [Ca2+]i trace is to show the [Ca2+]i change over time. To make these figures, we average the [Ca2+]i data over time from each individual experiment. However, the trace is not able to demonstrate statistical significance, and that is why we have a bar graph with each individual data points to demonstrate the statistical analysis. In Fig 6 g, the RvD1-H2 shows a reduction, but the p value is 0.07 according to our statistical analysis method (One-Way ANOVA). 

All lines on Fig. 5f look very similar.

Response II-5: Fig 5 is now fig 6. Similarly to the previous comment, to make these figures, we average the overtime [Ca2+]i data from each individual experiments. If the response from two sets of experiments is similar, it will be hard to separate them. In the revised figures, we used color to indicate each trace to make them more distinguishable. In addition we are analyzing peak [Ca2+]I that represent peak minus basal. As the basal values differ between experimental conditions it is difficult to see differences without analysis provided in the bar graphs.

No direct evidence of phosphorylation (activation) is provided. For example, experiments with [gamma-32P]-ATP may provide it.

Response II-6: We agree with the reviewer that we did not provide direct evidence of phosphorylation. We modified the text by changing phosphorylation to regulation by protein kinases in the manuscript.

Reviewer 3 Report

The aim of the presented manuscript was to analyse the interaction of histamine induced [Ca2+] increase and mucin secretion in conjunctival goblet cells with resolvin D2 and resolvin D1 and counter-activation  of protein kinases. Described research is a continuation of the Authors studies [e.g. FASEB J. 33, 8468–8478 (2019) and other] on the activity of resolvin D2 or D1 (RvD2 and RvD1) in different cellular models. In present work obtained were interesting results on different activity of  RvD2 and RvD1 using protein kinases to counter-regulation of histamine-induced [Ca2+]i increase induced by histamine H1-H4 receptors agonists. As a result was stated which type of kinases (PKA or PKC) takes part in the pathways of these signals transduction and which subtypes of histamine receptors are engaged.

Overall, the manuscript is well written, the introduction is very informative.  The conclusions are appropriate based on the manuscript as it is written, To enhance manuscript value please:

  • Add the figure with RvD1 and RvD2 structures and the way of their biosynthesis from docosahexaenoic acid
  • Try to make Figure 10 more clear, even enlarged description of proteins are difficult to read
  • Add list of explained abbreviations

Author Response

The aim of the presented manuscript was to analyse the interaction of histamine induced [Ca2+] increase and mucin secretion in conjunctival goblet cells with resolvin D2 and resolvin D1 and counter-activation of protein kinases. Described research is a continuation of the Authors studies [e.g. FASEB J. 33, 8468–8478 (2019) and other] on the activity of resolvin D2 or D1 (RvD2 and RvD1) in different cellular models. In present work obtained were interesting results on different activity of RvD2 and RvD1 using protein kinases to counter-regulation of histamine-induced [Ca2+]i increase induced by histamine H1-H4 receptors agonists. As a result was stated which type of kinases (PKA or PKC) takes part in the pathways of these signals transduction and which subtypes of histamine receptors are engaged.

Response III-1: We thank Reviewer 3 for recognizing the innovation of this work and the insightful recommendation. Each suggestion has been addressed in the manuscript revision, as detailed below: 

Overall, the manuscript is well written, the introduction is very informative.  The conclusions are appropriate based on the manuscript as it is written, To enhance manuscript value please:

Add the figure with RvD1 and RvD2 structures and the way of their biosynthesis from docosahexaenoic acid

Response III-2: We appreciate the suggestion. The biosynthetic pathway as well as the molecular structure has been added (see Figure 1).

 to make Figure 10 more clear, even enlarged description of proteins are difficult to read

Add list of explained abbreviations

Response III-3: Fig 10 is now fig 11. We changed the front size of the text and added the abbreviation list as requested (see Figure 11).

Round 2

Reviewer 2 Report

From this data, it is not evident that
"RvD2 and RvD1 each counter-regulates the action of histamine via different histamine receptor subtypes with RvD2 acting on H1, 3, and 4 and RvD1 acting on H1 and 3. Neither resolvin counter-regulates the H2 receptor subtype."

Using parametric statistics for these datasets looks quite odd, besides.
If we look at Fig. 6 E and F (which are not indicated) it is particularly strange that H3 is involved, while H2 is not. I do understand the reply, but if we look at Fig. 6G it is clear that the only experimental point may change all the results about H specificity. Hence, I do not feel I can review this MS anymore.